# Dynamic Contrast-Enhanced and Diffusion-Weighted Imaging in Magnetic Resonance in the Assessment of Peritoneal Recurrence of Ovarian Cancer in Patients with or Without BRCA Mutation

**DOI:** 10.3390/cancers16223738

**Published:** 2024-11-05

**Authors:** Melania Jankowska-Lombarska, Laretta Grabowska-Derlatka, Leszek Kraj, Pawel Derlatka

**Affiliations:** 1Second Department of Radiology, Medical University of Warsaw, Banacha 1a St., 02-097 Warsaw, Poland; melanie@vp.pl; 2Department of Oncology, Medical University of Warsaw, Banacha 1a St., 02-097 Warsaw, Poland; leszek.kraj@wum.edu.pl; 3Department of Molecular Biology, Institute of Genetics and Animal Biotechnology, Polish Academy of Science, 01-447 Magdalenka, Poland; 4Second Department of Obstetrics and Gynecology, Medical University of Warsaw, Karowa 2 St., 00-315 Warsaw, Poland; pawel.derlatka@wum.edu.pl

**Keywords:** ovarian cancer, recurrence, MRI, BRCA 1 or BRCA2 mutations, DCE, DWI

## Abstract

Recurrence in patients with ovarian cancer is common and causes difficulties for oncologists, radiologists and gynecologists. The mutation rate of the BRCA gene in high-grade serous ovarian cancer is greater than 21%. Patients suspected of having local recurrence and intraperitoneal spread are examined via magnetic resonance, owing to its excellent soft tissue resolution. The aim of this study was to determine the differences in diffusion-weighted imaging and dynamic contrast enhancement parameters in patients with peritoneal HGSOC recurrence with or without BRCA mutations. In this study, we revealed statistically significant differences in the DWI and DCE MRI parameters between patients with BRCA mutations and BRCA wild type. Additionally, we showed the difference between the parameters on MRI and the size of the peritoneal metastases. Adding DCE perfusion to the MRI protocol for detecting ovarian cancer recurrence in patients with BRCA mutations could be a valuable tool.

## 1. Introduction

Epithelial ovarian cancer (EOC) is associated with high morbidity and mortality because it is usually detected at an advanced stage [1,2]. The leading type of malignant tumor of the ovary is high-grade serous ovarian cancer (HGSOC) [3].

Approximately 15% of unselected cases of EOC, tubal, or peritoneal carcinoma are caused by germline mutations in the BRCA1 and BRCA2 genes. According to current large-scale studies of patients with HGSOC, the mutation rate in this type of cancer is estimated to be greater than 21%. Germline mutations in more recently identified ovarian cancer predisposition genes, such as RAD51C, RAD51D and BRIP1, are also observed in approximately 3% of patients [4].

Tumor prognosis depends on the use of optimal cytoreductive surgery and adjuvant platinum-based chemotherapy [5,6]. Current maintenance treatment with bevacizumab or polyadenosine diphosphate-ribose polymerase (PARP) inhibitors is associated with longer progression-free survival (PFS) [7,8,9,10,11].

A similar recurrence rate has been reported between BRCAwt and BRCAmut groups (35.3% vs. 29.2%) [12]. The basic method of treating recurrent ovarian cancer is chemotherapy. Secondary cytoreductive surgery is currently an option for second-line treatment in selected patients [13]. The choice of chemotherapy program depends primarily on the time of disease recurrence, the drugs used in the first line of treatment, and the molecular state of the tumor [14]. In addition, maintenance therapy with antiangiogenic therapy and PARP inhibitors has emerged as the standard of care. Novel combinations, including immunotherapy and antiangiogenic agents, have also been developed [15,16].

The role of imaging in the treatment of ovarian cancer recurrence is crucial. All clinically available methods, such as computed tomography (CT), magnetic resonance imaging (MRI), and positron emission tomography (PET) CT, are used, and each method has advantages and limitations. The main indications for MRI are suspicion of local recurrence in the pelvis and intraperitoneal spread [17,18]. Owing to its excellent high resolution of soft tissues, MRI allows for the distinction between post-treatment changes and tumor recurrence [18]. We know that intraperitoneal lesions, like primary tumors, differ in terms of MRI diffusion and perfusion parameters depending on the type of EOC, e.g., HGSOC vs. low-grade serous ovarian cancer (LGSOC) [19]. To date, there are no studies on the diffusion and perfusion parameters in recurrent ovarian cancer. However, there are reports that secondary and recurrent tumors are better vascularized [20]. Assessment of diffusion-weighted imaging (DWI) parameters, especially dynamic contrast enhancement (DCEs), in recurrent tumors may be helpful in selecting therapies for recurrent ovarian cancer that combine PARP inhibitors and antiangiogenic agents [20,21,22].

The aim of this study was to determine the differences in the DWI and DCE parameters in patients with peritoneal HGSOC recurrence in the BRCAmut and BRCAwt groups.

## 2. Materials and Methods

### 2.1. Study Protocol

This retrospective analysis of the abdominal and pelvic MR images of 43 patients aged 21–76 years was conducted in the Second Department of Radiology at the Medical University of Warsaw. The patients who were examined were suspected of having ovarian cancer recurrence after first-line treatment, and 18 of these patients had BRCA1/BRCA2 mutations. Relapse was suspected due to an elevated Ca-125 (above twice the nadir value) and computed tomography findings. The study group included only patients with platinum-sensitive HGSOC [23,24].

### 2.2. Imaging Protocol

All of the patients underwent MRI on a 1.5 Tesla MR scanner (MAGNETOM Avanto, Siemens AG, Erlangen, Germany). The imaging protocol included the following: T1 in and out phase, T1 pre- and postcontrast (dynamic contrast-enhanced; DCE), T2 turbo spin echo (TSE), fat-suppressed T2, T2 weighed sequence, turbo inversion recovery magnitude (TIRM), and diffusion-weighted imaging (DWI). For the DCE sequences, patients were administered a bolus dose (0.1 mmol/kg) of gadobutrol (Gadovist, Bayer AG, Leverkusen, Germany) followed by a bolus dose of 20 mL of 0.9% saline solution. Motion correction was applied automatically. Several parameters are shown in Table 1. DWI was performed using the b values of 0, 50, 400 and 1200 mm^2^/s. Apparent diffusion coefficient (ADC) maps were produced automatically via computer software.

Two radiologists with experience in abdominal and pelvic MRI (one with more than 15 years of experience in oncological imaging and one with a European diploma in radiology) independently analyzed the images. First, the abdomen was divided into the following areas according to the PCI index: the right hypochondrium, epigastrium, left hypochondrium, left flank, left iliac fossa, pelvis, right iliac fossa, central, right flank, proximal jejunum and ileum and distal jejunum and ileum. Each area was analyzed according to the presence of peritoneal metastases and their size, which were further subdivided into groups: small (<1 cm), medium (1–2 cm) and large (>2 cm). Then, the metastasis with the qualitatively highest diffusion restriction was selected to quantitatively measure the ADC values. Small ROIs (5 mm) were placed into the areas with the solid part of the metastasis on the DWI scans, which were acquired with a b value of 1500 mm^2^/s, and the cystic parts and artifacts were avoided. Then, according to the DWI selection, the ROIs were placed on the ADC maps (Figure 1 and Figure 2). The measurements on the ADC maps were saved. Three measurements were made by each radiologist independently.

For the DCE T1 images, first, precontrast images were acquired, and then each phase was subsequently acquired after contrast was administered. The perfusion maps and charts were produced automatically via AV Server 4.2 software. The areas of metastases were analyzed as described above. For the peritoneal metastases selected for measuring the ADC values, small ROIs were placed on the T1 postcontrast images and recreated on the DCE parametric maps (Figure 3 and Figure 4). Again, three measurements were made by each radiologist independently. Parameters such as the time to peak (TTP) and perfusion maximum enhancement (Perf. Max. En.) were measured, and all the values were saved.

### 2.3. Statistical Analysis

IBM SPSS Statistics (version 28.0.1.0 (142)) was used to analyze the distribution of the variables, perform some of the statistical tests, and calculate some of the statistics. PQStat (version: 1.8.6.102.) was used for the same purpose: to analyze the data and to prepare the visualizations included in this article. Power BI (version: 2.123.684.0) was used to create the chart showing the percentage distribution of the peritoneal metastasis size between the patients with and without BRCA mutations. The Shapiro-Wilk test was used to determine whether a variable was normally distributed. All numerical values characterized by a normal distribution are presented as the means and standard deviations, whereas the qualitative variables are presented as raw numbers and percentages. The *p*-values were determined via three different tests, and a *p*-value less than 0.05 indicated a statistically significant difference. For interval scale variables that met the condition of a normal distribution and for those that were independent, the t test for independent groups was used, which is specifically designed for such conditions. For normally distributed quantitative variables, one-way ANOVA for independent groups was used. For nominal and independent variables, the chi-square test or Fisher’s exact test was used. For variables characterized by a normal distribution, the post hoc Fisher’s LSD test was performed to determine statistically significant differences between groups. For interval variables, ICC was used to determine the interobserver agreement between the two reviewers.

## 3. Results

### 3.1. Statistical Analysis of the Selected Parameters

#### 3.1.1. Categorizing Patients According to BRCA Mutation Status

There were 43 women in the patient cohort with an average age of approximately 56 years.

All patients underwent surgery. Twenty-five patients with FIGO stage III underwent primary cytoreductive surgery, including 19 who completed surgery with no residual macroscopic disease (R0 surgery). Six had residual disease < 10 mm (R1).

Eleven patients started treatment with neoadjuvant chemotherapy followed by interval debulking surgery. Ten patients had no residual disease (R0), and one had >10 mm residual disease (R2). The clinicopathological characteristics of the study group are shown in Table 2.

The analyzed parameters were the ADC, TTP and Perf. Max. En. and the size of the peritoneal implants. The patients were categorized according to the absence (*N* = 25, 58%) or presence (*N* = 18, 42%) of BRCA mutations. The obtained results are shown in Table 3.

The ADC levels were significantly different between the two groups of patients. The mean ADC in the without (*w*/*o*) mutation group of patients was 788.7 (*SD*: 139.5), whereas for the with mutation group, this value was 977.3 (*SD*: 103; *p* = 0.00002). Regarding the TTP and Perf. Max. En. parameters, statistically significant differences were also observed between the two groups; for both parameters, the *p*-value was <0.01. The average TTP for patients without mutations was 256.3 (*SD*: 50), whereas for patients with mutations, the average TTP was 160.6 (*SD*: 35.5). For the Perf. Max. En. parameter, a lower average value of 148.6 (*SD*: 12.3) was observed in the group without mutations, whereas for patients with mutations, this value was 233.6 (*SD*: 29.2).

On average, patients with BRCA mutations were characterized by greater ADC and Perf. Max. En. values, whereas the TTP value was lower in patients with BRCA mutations than in patients without BRCA mutations (*p*-value < 0.05).

#### 3.1.2. Patient Categorization According to the Size of the Peritoneal Metastasis

In addition to categorizing patients according to the presence of BRCA mutations, patients were also categorized on the basis of the size of the peritoneal metastasis.

For further analysis, the patients were assigned to group one (peritoneal metastasis < 1 cm, *N* = 18, 42%), group two (peritoneal metastasis between 1 and 2 cm, *N* = 17, 40%), or group three (peritoneal metastasis > 2 cm, *N* = 8, 18%). The results are shown in Table 4.

Statistically significant differences were noted for the ADC parameter among the three groups. There were significant differences in the mean ADC between patients in group two (mean ADC = 760.5, *SD*: 108) and the combined group comprising patients in group one (mean ADC = 965.6, *SD*: 108.7) and those in group three (mean ADC = 895.3, *SD*: 203; *p* = 0.0003). There were no statistically significant differences in the mean ADC between the patients in group one and those in group three; i.e., patients in group one and group three were not significantly different from each other in terms of the mean ADC.

### 3.2. Interobserver Agreement

The intraclass correlation coefficient (ICC) was used to examine the interobserver agreement for the ADC and TTP quantitative parameters. The ICCs revealed excellent, statistically significant interobserver agreement between the two observers for both parameters.

The interobserver concordance oscillated at the level of excellent concordance, and the ICC was >0.9 (Table 5).

## 4. Discussion

Our study involving 43 patients revealed a relationship between the DWI and DCE MRI parameters and BRCA1/2 mutation status in molecular studies of recurrent HGSOC patients. The mean ADC in patients with BRCAwt was lower than that in patients with BRCAmut: 788.7 vs. 977.3, *p*-value = 0.00002. The average TTP value for patients with BRCAwt was greater than that for patients with mutations: 256.3 vs. 160.6, *p*-value < 0.01. The Perf. Max. En. value was lower in the BRCAwt group: 148.6 (*SD*: 12.3) vs. 233.6 (*SD*: 29.2), *p*-value < 0.01.

Similar protocols of studies using ADC and selected perfusion parameters in 1.5 T and 3.0 T MRI have been described in previous publications [25,26,27].

The ADC values, which are derived from DWI and are inversely correlated with tumor cellularity, were greater in the BRCAmut group than in the BRCAwt group (*p* < 0.00002). The decrease in restricted diffusion, which was statistically significant in the tumors of patients with BRCAmut, was more fluid. Higher ADC values in peritoneal metastases before treatment may be linked to a poorer response to chemotherapy and a poorer prognosis [28].

DCE parameters allow for the evaluation of tumor vascularization/angiogenesis. Our study of recurrent tumors revealed significantly greater TTP values (longer time to peak) in the BRCAwt group than in the BRCAmut group (*p* < 0.01). The Perf. Max. En. parameter showed an inverse correlation, and the values were significantly greater in the BRCAmut group. This leads to the conclusion that peritoneal metastases in tumors with BRCA 1/2 mutations are more vascularized.

We also examined the correlation between the size of the peritoneal metastases and the DWI and DCE parameters. The results revealed that the highest diffusion restriction, in the form of the lowest ADC values, was observed in the medium-sized metastases (1–2 cm). Metastases that were <1 cm or >2 cm did not differ significantly in terms of the ADC values; however, their values were greater. There was no correlation between the TTP or the Perf. Max. En. parameters and the size of the peritoneal implants.

Finally, we determined the correlations between the presence of the BRCA1/2 mutations, the size of the recurrent implants and the DWI or DCE parameters. Our results revealed that the TTP and Perf. Max. En. parameters are dependent only on the presence or absence of BRCA mutations and not on tumor size. For the ADC values, the correlation was more complex and may require further study.

DCE parameters are widely used to characterize tumors on MRI or to evaluate the response to treatment [29]. Low perfusion values (Ktrans and kep) are inversely correlated with the risk of tumor recurrence in patients with HGSOC [30]. The TTP is the time at which the contrast agent reaches its maximal concentration, and maximal enhancement is the peak contrast concentration, which is why tumors with high vascularization will have low TTP values [31] and high perfusion maximum enhancement values on DCE [32].

Studies have shown that higher ADC values are associated with high VEGF protein expression in metastatic endothelial cells and that lower TTP values are associated with more aggressive tumors [33]. Additionally, lower TTP values were associated with shorter recurrence-free survival [34].

VEGF acts as a proangiogenic agent. In other tumors, such as gastric cancer, a positive predictive value has been demonstrated between DCE parameters and VEGF receptor expression [35], whereas studies examining the correlation between DCE parameters and VEGF in ovarian cancer have shown an inverse correlation [36]. One possible explanation is that the blood flow is already increased in ovarian cancer and that high VEGF expression is no longer needed [36]. There are few studies examining peritoneal metastases on MRI in patients with EOC; however, one of the studies compared primary tumors and metastasis and demonstrated that ovarian cancer metastases show increased expression of proangiogenic proteins compared with primary tumors [37].

DWI parameters are also correlated with VEGF expression. A negative correlation between the ADC value and VEGF expression was observed in primary EOC tumors [38]. Lindgren and coauthors reported a positive correlation in one of the first articles examining the correlation between ADC values and VEGF expression in ovarian cancer. The difference in the results may be because, in one study, the ADC values were obtained only from the primary tumor, whereas in the second study, they were obtained from the primary and secondary tumors (metastases). Lidgren also reported that VEGF expression is greater in metastases than in primary tumors [19]. An inverse correlation between the ADC value and VEGF expression was also found in prostate cancer [39] and breast cancer [40].

In our study, we observed the lowest ADC values in the group with peritoneal metastases between 1 cm and 2 cm, and the difference was statistically significant compared with both the <1 cm group and the >2 cm group. The highest ADC values were found in the group with the smallest metastases (<1 cm). We suspect that the difference in our study is due to the partial volume effect, which occurs when there is more than one type of tissue in the voxel.

Unfortunately, studies examining the ADC values and the size of the peritoneal deposits are limited. A possible indirect reference here may be studies of DWI in lymph nodes less than 1 cm showing significant differences in the ADC between metastatic and normal nodes. The authors suggested that such a small size of the node excludes the presence of necrosis [41]. Some authors have suggested that the partial volume effect does not significantly influence the measurements, even in lymph nodes of approximately 6 mm [42]; however, other studies have suggested that to avoid the partial volume effect, the mass size should be at least twice as large as the slice thickness of the DWI sequence [43]. In our MRI protocol, the slice thickness on DWI was 6 mm; however, when peritoneal deposits were observed on the slice images, we also examined them because of the limited population group. Moreover, placing the ROI in a small lesion is problematic, and taking the size and ROI shape into account while measuring the ADC value is crucial [44,45]. The values obtained from placing a small ROI in the darkest area of the ADC map rather than in the entire lesion show better sensitivity, specificity and positive predictive value [46]. Our results revealed that there was a significant correlation only between the group with 1–2 cm peritoneal deposits in the corresponding BRCAwt and BRCAmut groups, possibly because in these groups, the ROIs were small, the lesions were homogenous, and we could avoid areas of micronecrosis.

A recent MRI study of EOC primary tumors in BRCAwt and BRCAmut patients revealed no differences in the ADC and DCE parameters [37], unlike our findings. This difference may be because we examined peritoneal metastases instead of primary tumors, and as mentioned above, some authors have proven that secondary tumors are better vascularized. Furthermore, the study revealed that BRCA1mut patients had more enhanced tumors than the BRCA2mut group did; thus, the study group also differed, as we did not examine the differences in the MRI characteristics of the patients with each mutation type.

The use of diffusion parameters and ADC maps is a recognized method for the diagnosis of cancer recurrence. In our study, we showed that in the case of ovarian cancer recurrence, ADC values have certain limitations related to the size of the metastases, whereas perfusion parameters (TTP and max enh) do not depend on the size of the metastases and, in our opinion, have greater diagnostic value in tumors that are richly vascularized. To reliably compare the ADC values, authors should select peritoneal metastases not only with the qualitatively greatest diffusion restriction but also with similar sizes (e.g., <2 cm). These findings may be used to qualify patients for further treatment with a combination of PARP inhibitors and antiangiogenic agents [21,22].

In the future, machine learning may be helpful in differentiating these lesions, as it provides excellent accuracy that surpasses conventional analysis and may improve the assessment of ovarian cancer recurrence in the peritoneum [47].

One limitation of our study was that it was a single-center study with a small sample size, which is why we also combined patients with BRCA1 and BRCA2 mutations. Therefore, maintenance therapy differed between the two groups, which could have affected the results. Although we controlled for ROI placement, chose a small ROI, and achieved very good interobserver agreement, we could not avoid the partial volume effect in the smallest peritoneal metastases. Additionally, we examined recurrent tumors, which is why further studies are needed to determine whether there is a similar correlation in newly diagnosed patients.

## 5. Conclusions

To conclude, adding DCE perfusion to the MRI protocol for detecting ovarian cancer recurrence in patients with BRCA mutations could be a valuable tool regardless of the size of the metastases.

## Figures and Tables

**Figure 1 cancers-16-03738-f001:**
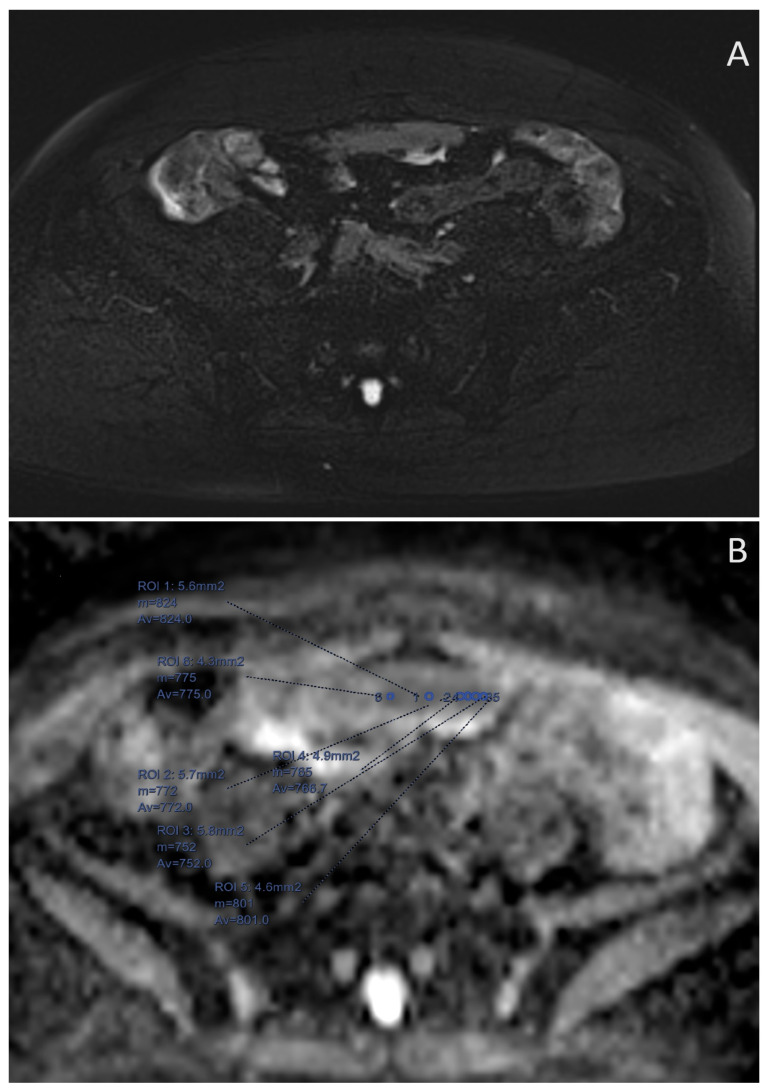
MRI of a 65-year-old patient with HGSOC recurrence BRCAwt. (**A**) Midline large peritoneal metastases on the STIR sequence. (Short tau inversion recovery) (**B**) ADC map image of the same peritoneal implant with small ROIs placed in the areas that qualitatively have the lowest signal.

**Figure 2 cancers-16-03738-f002:**
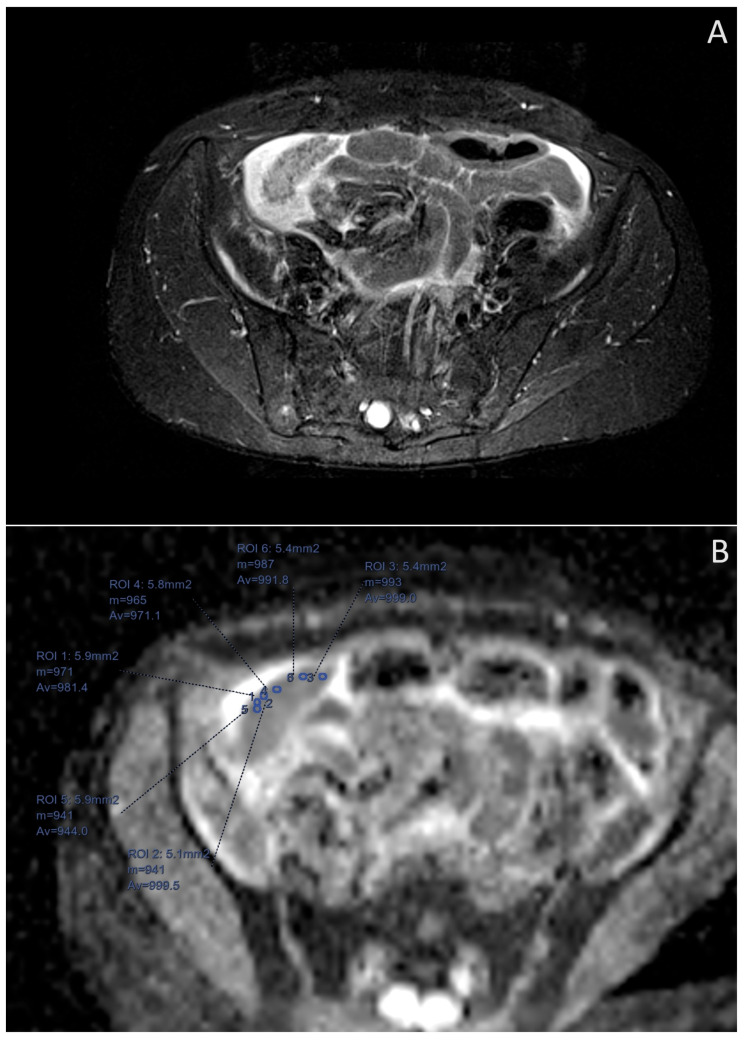
Images of the same 46-year-old patient with HGSOC recurrence BRCAmut. (**A**) Large peritoneal metastases in the right iliac fossa on the STIR sequence. (**B**) ADC map. Magnified image of the same peritoneal implant with small ROIs placed in the areas that qualitatively have the lowest signal.

**Figure 3 cancers-16-03738-f003:**
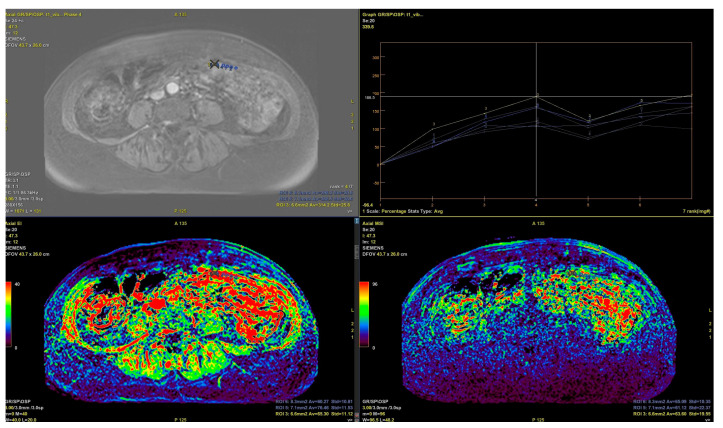
Images from the 65-year-old patient with HGSOC recurrence BRCAwt, contrast enhancement maps and curves.

**Figure 4 cancers-16-03738-f004:**
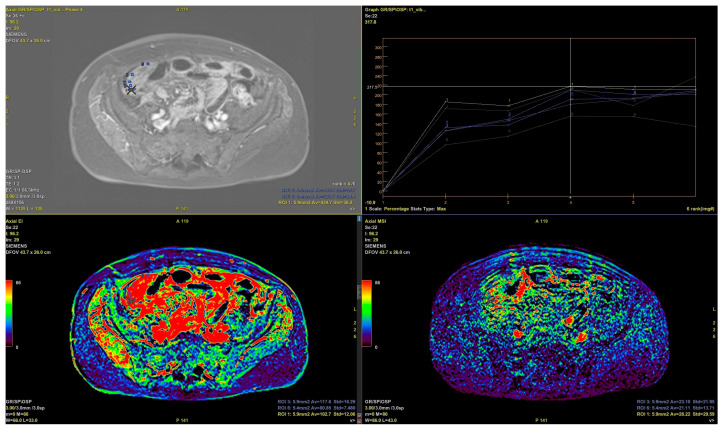
MRI of a 46-year-old patient with HGSOC recurrence with a BRCAmut. Contrast en-hancement maps and curves.

**Table 1 cancers-16-03738-t001:** Selected parameters used in the study protocol.

Sequence	TE	TR	Flip Angle	Slice Thickness (mm)	FOV (mm)	Orientation	NEX	Breath Hold
T1 in and out phase	1–2.22, 2–4.92	125	70	6	300 × 225	axial	1	no
T2 TIRM	89	4840	150	3	219 × 199	axial	2	no
T2 TSE	116	4250	137	3	299 × 286	axial, coronal and sagittal	1	no
T2 fat sat Tra	123	2110	150	3	300 × 300	axial	1	no
DWI	76	3200	90	6	289 × 234	axial	2	no
Vibe T1 GRE	1.1	3.1	10	3	288 *×* 156	axial, sagittal and coronal	1	no

**Table 2 cancers-16-03738-t002:** Clinicopathological characteristics of the 43 patients included in this study.

Variable	BRCAmut (*n* = 18)	BRCAwt (*n* = 25) Including 5 HRD Positive	*p*-Value
Age (avg ± sd)	56.4 ± 9	56.6 ± 12.3	0.9 ^t^
FIGO stage			0.98 ^F^
I	2	3	
II	1	1	
III	14	19	
IV	1	2	
First-line chemotherapy			0.22 ^F^
PCL + CBDCA	18	23	
CBDCA mono	0	2	
Maintenance treatment			0.25 ^F^
Bevacizumab	0	5	
PARP inhibitors	14 (olaparib)	15 (niraparib)	
BEV & PARPi	1	1	
other	3	4	
Treatment response			0.47 ^F^
Complete response	17	22	
Partial response	1	3	

The values are expressed as raw numbers and percentages. Test of significance: *p*-value: t—*t* test for independent groups; F—chi-square or Fisher’s exact test. Abbreviations: HRD—homologous recombination deficiency; PCL—paclitaxel; CBDCA—carboplatin; mono-monotherapy; PARP—polyadenosine diphosphate-ribose polymerase; BEV—bevacizumab; PARPi—polyadenosine diphosphate-ribose polymerase inhibitors.

**Table 3 cancers-16-03738-t003:** Statistical analysis of the parameters (ADC, TTP and Perf. Max. En.) and age of the two groups (patients with or without (*w*/*o*) BRCA mutations).

Characteristic	BRCAwt (*N* = 25)	BRCAmut (*N* = 18)	All Patients (*N* = 43)	*p*-Value
ADC	788.7 ± 139.5	977.3 ± 103	867.7 ± 155.8	<0.001 ^t^
TTP	256.3 ± 50	160 ± 35.5	216.2 ± 65	<0.01 ^t^
Perf. Max. En.	148.6 ± 22.4	233.6 ± 29.2	184.2 ± 49.3	<0.01 ^t^
Size of the implants ^a^				0.01 ^F^
<1 cm, n (%)	6 (24%)	12 (67%)	18 (41%)	
>1 cm, <2 cm n (%)	14 (56%)	3 (17%)	17 (40%)	
>2 cm, n (%)	5 (20%)	3 (17%)	8 (19%)	

The values are expressed as the arithmetic mean with the standard deviation. ^a^ The values are expressed as raw numbers and percentages. Test of significance: *p*-value: t—*t* test for independent groups; F—chi-square or Fisher’s exact test.

**Table 4 cancers-16-03738-t004:** Statistical analysis of the parameters (ADC, TTP and Perf. Max. En.) in the three groups of patients stratified according to the size of the peritoneal metastasis.

	Size of the Peritoneal Metastasis				
Characteristic	<1 cm (*N* = 18)	>1 cm, <2 cm (*N* = 17)	>2 cm (*N* = 8)	All Patients (*N* = 43)	*p*-Value
ADC	956.6 ± 108.7	760.5 ± 108	895.3 ± 203	867.7 ± 155.8	0.0003
*Pairwise comparison*	B	A	B		
TTP	203 ± 67.2	234.4 ± 57.5	207.6 ± 73.8	216.2 ± 65	0.3
Perf. Max. En.	196.3 ± 50.9	167.5 ± 48.2	192.5 ± 43	184.2 ± 49.3	0.2

The values are expressed as the arithmetic mean with the standard deviation. Test of significance: *p*-value: one-way ANOVA for independent groups. Pairwise comparisons (post hoc Fisher’s LSD); same letters = insignificant difference; different letters = significant difference. Abbreviations—see Table 3.

**Table 5 cancers-16-03738-t005:** Intraclass correlation coefficients of the ADC and TTP parameters as determined by the two observers.

Parameter	ICC	95% CI	*p*-Value
ADC	0.98	0.96–0.99	<0.01
TTP	0.95	0.91–0.97	<0.01

ICC: intraclass correlation coefficient; CI: confidence interval.

## Data Availability

The raw data supporting the conclusions of this article will be made available by the authors upon request.

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
