# Peer review of "Dynamic Contrast-Enhanced and Diffusion-Weighted Imaging in Magnetic Resonance in the Assessment of Peritoneal Recurrence of Ovarian Cancer in Patients with or Without BRCA Mutation"

_cancers, 2024, doi:10.3390/cancers16223738_

Round 1

Reviewer 1 Report

Comments and Suggestions for Authors

I read with interest the manuscript entitled "DCE and DWI magnetic resonance imaging in the assessment of peritoneal recurrence of ovarian cancer in patients with or without BRCA mutation"

I suggest adding the full name of the abbreviation to the title.

The introduction is clearly and concisely written with a clear statement of the aim of the study.

Please include the reference for the above "Relapse was suspected due to elevated Ca-125 (above twice the nadir value) and computed tomography findings."

Within Table 1, it is unclear where you wrote the medians. Please write the medians in the appropriate place in the table. Table 1 should be part of Results, not Materials and Methods. It is also necessary to state the p-values ​​between the observed groups.

Under tables, it is necessary to list all used abbreviations, as well as their full names.

Please provide references for applied analyzes and measurement protocols.

Figures within the manuscript must appear in the order in which they are listed.

In tables, next to the p-values, indicate with a superscript which test was used.

All results that are listed in the text or tables do not need to be displayed via box-plots. Please remove the box-plots. Show the results in a table.

In relation to table 6, multiple dot graphs are unnecessary. Remove them.

Within the discussion, be careful in using the term „correlation“. Correlation is a statistical term! At the beginning of the discussion, briefly present the most relevant results in the form in which you calculated them.

Have you recognized all the limitations of your study? Think again.

Please search the databases, and you will notice that there are more articles on the given topic that you did not include in the discussion.

In conclusion section, stick strictly to your results!

In conclusion, revise the manuscript according to the instructions, clearly adhering to the structure of the article, not mixing parts of the results with the methodology and vice versa. Be careful in using terms, especially the term "correlation". Supplement the discussion with all relevant knowledge on the given topic, and in the conclusion stick strictly to your results, without generalizing.

Comments on the Quality of English Language

Moderate editing of English language required.

Author Response

Reply to Reviewer 1

I suggest adding the full name of the abbreviation to the title.

Thank you for the remark, title has been changed.

Please include the reference for the above "Relapse was suspected due to elevated Ca-125 (above twice the nadir value) and computed tomography findings."

Thank you for the remark, references has been included

Du K, Li Q, Huang J, Chan DW, Li J, Chang X, Wang H, Tang J, Yang Q. An increase of serum CA-125 to two times of nadir level strongly predicts the image-identified relapse of serous ovarian cancer. Sci Rep. 2024 Jul 1;14(1):14986.

Piatek S, Panek G, Lewandowski Z, Bidzinski M, Piatek D, Kosinski P, Wielgos M. Rising serum CA-125 levels within the normal range is strongly associated recurrence risk and survival of ovarian cancer. J Ovarian Res. 2020 Sep 2;13(1):102.

Within Table 1, it is unclear where you wrote the medians. Please write the medians in the appropriate place in the table.

Thank you for the remark, it has been changed.

 Table 1 should be part of Results, not Materials and Methods.

            Thank you for the remark, it has been changed.

 It is also necessary to state the p-values ​​between the observed groups.

Thank you for the remark, p-values have been added.

Under tables, it is necessary to list all used abbreviations, as well as their full names.

            Thank you for the remark, it has been changed.

Please provide references for applied analyzes and measurement protocols.

Thank you for the remark. Similar protocols of studies using ADC and selected perfusion parameters in 1.5T and 3.0T MRI have been described in previous publications, both ours and in studies of other authors. Some of them are already cited in this publication. 19,23,29,33

Below we present the next ones and add the appropriate paragraph in the discussion

Grabowska-Derlatka L, Derlatka P, Szeszkowski W, Cieszanowski A. Diffusion-Weighted Imaging of Small Peritoneal Implants in "Potentially" Early-Stage Ovarian Cancer. Biomed Res Int. 2016;2016:9254742. doi: 10.1155/2016/9254742. 

Halaburda-Rola, M.; Grabowska-Derlatka, L.; Kraj, L.; Stec, R.; Derlatka, P. Evaluation of Selected MRI Parameters in the Differentiation of Mucinous Ovarian Carcinomas and Metastatic Ovarian Tumors. Cancers 2024, 16, 3569. 

Li, H.M.; Zhang, R.; Gu, W.Y.; Zhao, S.H.; Lu, N.; Zhang, G.F.; Peng, W.J.; Qiang, J.W. Whole solid tumour volume histogram analysis of the apparent diffusion coefficient for differentiating high-grade from low-grade serous ovarian carcinoma: Correlation with Ki-67 proliferation status. Clin. Radiol. 2019, 74, 918–925. 

Figures within the manuscript must appear in the order in which they are listed.

Thank you for the remark, it has been changed.

In tables, next to the p-values, indicate with a superscript which test was used.

Thank you for the remark, appropriate descriptions have been added to the tables.

All results that are listed in the text or tables do not need to be displayed via box-plots. Please remove the box-plots. Show the results in a table.

            Thank you for the remark, they have been removed.

In relation to table 6, multiple dot graphs are unnecessary. Remove them.

            Thank you for the remark, they have been removed.

Within the discussion, be careful in using the term „correlation“. Correlation is a statistical term!

            Thank you for the remark, it has  changed, where there were no obvious statistical relationships

At the beginning of the discussion, briefly present the most relevant results in the form in which you calculated them.

            Thank you for the remark, it has  been presented.

Have you recognized all the limitations of your study? Think again.

Study limitations described in the discussion:  1. a single-center study with a small sample size, which is why we also combined patients with BRCA1 and BRCA2 mutations;

  1. maintenance therapy differed between the two groups, could have affected the results;
  2. the partial volume effect in the smallest peritoneal metastases due to the location of small ROIs
  3. the study only concerned recurrent disease.

Please provide any further suggestions

Please search the databases, and you will notice that there are more articles on the given topic that you did not include in the discussion.

            Due to previous corrections, we have added several items

In conclusion section, stick strictly to your results!

Thank you for your suggestion. We have corrected the paragraph

In conclusion, revise the manuscript according to the instructions, clearly adhering to the structure of the article, not mixing parts of the results with the methodology and vice versa. Be careful in using terms, especially the term "correlation". Supplement the discussion with all relevant knowledge on the given topic, and in the conclusion stick strictly to your results, without generalizing.

Thank you for your your valuable comments

Reviewer 2 Report

Comments and Suggestions for Authors

After reviewing the manuscript, I find it difficult to identify a significant advancement that would justify publishing this paper in the journal.

-The study has limitations, including small sample sizes, which hinder the ability to draw robust conclusions. Also, other limitations are listed correctly at the end of section 4.

- Considering the previously mentioned limitations, how could the presented result be integrated into a new protocol?

-Additionally, similar studies have already explored correlations between MRI parameters and ovarian cancer patients. What is the significance of this research compared to similar studies mentioned in the fourth section of the manuscript?

- There are typos throughout the manuscript. Please, corrections are needed.

In my opinion, there is room for improvement in this manuscript. After that, it may be reconsidered for publication in the journal.

Comments on the Quality of English Language

Please, see the report.

Author Response

Reply to Reviewer 2

After reviewing the manuscript, I find it difficult to identify a significant advancement that would justify publishing this paper in the journal. 

Thank you for your comments. Previously published works on the assessment of peritoneal implants in ovarian cancer focused on the assessment of changes before primary cytoreductive surgery or before neoadjuvant chemotherapy. Our goal was to draw attention to the possibilities offered by diffusion and perfusion parameters only in the case of diagnostic difficulties in recurrent ovarian cancer. Our attention, during the assessment of studies, was drawn primarily to patients with BRCA 1 mutations, where perfusion parameters in implants are significantly different than in patients without mutations. Therefore, we wanted to draw the attention of future researchers and oncologists to this group of patients and to the emphasis on diagnostics focused on quantitative assessment in MRI studies. And this, in our opinion, is a new idea.

-The study has limitations, including small sample sizes, which hinder the ability to draw robust conclusions. Also, other limitations are listed correctly at the end of section 4. 

We agree with the reviewer's opinion and therefore limit the conclusions to only those that strictly relate to our results

- Considering the previously mentioned limitations, how could the presented result be integrated into a new protocol?

Of course, the use of this technique requires further studies to confirm what we are working on, but we already believe that it is valuable. The protocol we use is similar to the protocol for assessing preoperative changes, similar protocols are also used by other centers, so searching for ovarian cancer recurrence in MRI is a reliable and objective assessment.

Previously published works mostly concerned the differential diagnosis of primary tumors, which we write about in the discussion, while in our work we focused on showing differences in implants in case of disease recurrence in patients in whom other methods did not provide a clear answer about disease progression. Our results showed that perfusion parameters such as TTP and Perf. Max. En. depend only on the presence or absence of BRCA mutations, and not on the size of the tumor, and this is valuable information for future researchers to assess recurrence also regarding perfusion parameters-

- There are typos throughout the manuscript. Please, corrections are needed.

Thank you for the remark, it has been changed.

In my opinion, there is room for improvement in this manuscript. After that, it may be reconsidered for publication in the journal.

Based on the reviewers' comments, we have made the suggested corrections, we hope that the corrected version of the work will be suitable for publication. Thank you.

Reviewer 3 Report

Comments and Suggestions for Authors

This study was retrospectively designed to evaluate the role of magnetic resonance imaging (MRI) in women with suspected recurrence of high-grade serous ovarian cancers and examined the differences in diffusion-weighted imaging (DWI) and dynamic contrast enhancement (DCE) parameters in patients with peritoneal relapsed disease with or without BRCA mutations.

The paper is well written and the English language is appropriate and understandable.

The clinical topics are interesting. To date, different imaging methods are available to evaluate suspected recurrence without clear advantages and limitations, and there are no studies on diffusion and perfusion parameters in recurrent ovarian cancer. Furthermore, there is an ever-increasing interest in evaluating the different clinical characteristics and outcomes of BRCA carriers with a significant and helpful role in selecting treatments for different settings of recurrent ovarian cancers.

This paper showed statistically significant differences in the DWI and DCE MRI parameters in patients with BRCA mutations (peritoneal metastases more vascularized) in comparison to those with BRCA wild type. Additionally, the Authors revealed that MRI parameters were dependent only on the BRCA status not on tumor size.

However, the conclusions are appropriate: DCE perfusion to the MRI protocol for detecting ovarian cancer recurrence in patients with BRCA mutations could be a valuable tool.

The strengths and limitations of this analysis are reported carefully. To date, only a few studies have evaluated the relationship between BRCA status and MRI parameters in relapsed ovarian cancer. Unfortunately, the sample size of the patients included in this analysis is very small.

The cited references are mostly recent and include relevant publications.

Specific comments:

Could the authors provide more details on patient characteristics including the surgical treatment (primary cytoreduction vs interval debulking surgery after neo-adjuvant chemotherapy), residual tumour at the end of primary surgery, HRD status, and the median time to relapse?

Table 1 must be revised regarding:  Median [48-52]

Peritoneal implants (row 130) must be corrected.

Author Response

Reply to Reviewer 3

Could the authors provide more details on patient characteristics, including surgical treatment (primary cytoreduction vs. interval debulking after neoadjuvant chemotherapy), residual tumor at the end of primary surgery, HRD status, and median time to relapse?

 Thank you for the remark, paragraph describing this data has been added.

Table 1 needs to be amended with respect to: Median [48-52]

 Thank you for the remark, it has been changed.

Peritoneal implants (row 130) need to be corrected.

Thank you for the remark, it has been changed

Reviewer 4 Report

Comments and Suggestions for Authors

In this submission to Cancers, the authors determine the differences in diffusion-weighted imaging and dynamic contrast enhancement parameters in patients with peritoneal HGSOC recurrence with or without BRCA mutations. The authors revealed statistically significant differences in the DWI and DCE MRI parameters between patients with BRCA mutations and BRCA wild type. The authors also showed the difference between the parameters on MRI and the size of the peritoneal metastases. The authors conclude that adding DCE perfusion to the MRI protocol for detecting ovarian cancer recurrence in patients with BRCA mutations could be a valuable tool.

I find this manuscript to be of interest to cancer researchers as well as readers of this journal. As such, I am generally supportive of publication with a few minor edits. Specifically, there has been prior work on using machine learning for health aspects, which should be noted: Sci. Rep. 2023, 13, 10478 and Environ. Sci. Technol. Lett. 2023, 10, 1017–1022. Specifically, these prior works used machine learning approaches to help inform diagnostics and experiments (particularly in the area of BRCA and other health contaminants). I do recognize that the main point of this paper is not to obtain the most accurate simulations of health effects, so I am not asking the authors to do machine-learning calculations. However, the authors should note that these prior studies of machine learning for health effects are synergistic with the experiments used here.

Author Response

Reply to Reviewer 4.

Thank you, your comment is valuable and therefore we have added a paragraph on machine learning to the discussion.

Round 2

Reviewer 2 Report

Comments and Suggestions for Authors

I am satisfied with the authors' response and recommend the manuscript for publication in this journal.